# Influence of Dissolving Fe–Nb–B–Dy Alloys in Zirconium on Phase Structure, Microstructure and Magnetic Properties

**DOI:** 10.3390/ma14102526

**Published:** 2021-05-12

**Authors:** Grzegorz Ziółkowski, Artur Chrobak, Ewa Talik, Joanna Klimontko, Dariusz Chrobak

**Affiliations:** 1Institute of Physics, University of Silesia in Katowice, 75 Pułku Piechoty 1A, 41-500 Chorzów, Poland; grzegorz.ziolkowski@us.edu.pl (G.Z.); ewa.talik@us.edu.pl (E.T.); joanna.klimontko@us.edu.pl (J.K.); 2Institute of Materials Engineering, University of Silesia in Katowice, 75 Pułku Piechoty 1A, 41-500 Chorzów, Poland; dariusz.chrobak@us.edu.pl

**Keywords:** hard magnetic materials, rare earth alloys, vacuum mold suction casting

## Abstract

This paper refers to the structural and magnetic properties of [(Fe_80_Nb_6_B_14_)_0.88_Dy_0.12_]_1−x_Zr_x_ (*x* = 0; 0.01; 0.02; 0.05; 0.1; 0.2; 0.3; 0.5) alloys obtained by the vacuum mold suction casting method. The analysis of the phase contribution indicated a change in the compositions of the alloys. For *x* < 0.05, occurrence of the dominant Dy_2_Fe_14_B phase was observed, while a further increase in the Zr content led to the increasing contribution of the Fe–Zr compounds and, simultaneously, separation of crystalline Dy. The dilution of (Fe_80_Nb_6_B_14_)_0.88_Dy_0.12_ in Zr strongly influenced the magnetization processes of the examined alloys. Generally, with the increasing *x* parameter, we observed a decrease in coercivity; however, the unexpected increase in magnetic saturation and remanence for *x* = 0.2 and *x* = 0.3 was shown and discussed.

## 1. Introduction

Searching for new magnetic materials is still of great importance, considering the continuing demand for their unique characteristics, which are required for a variety of applications [1,2,3,4,5]. This work refers to the so-called hard magnetic materials that are widely used in, for example, automotive or electric technologies. The main problem, as well as research direction, is to fill the gap between classical (e.g., ALNICO alloys [6,7]) and rare earth (RE) permanent magnets (e.g., Nd–Fe–B types of alloys [8,9,10]) by new magnetic systems without or with reduced RE content [11,12]. It seems that one possible way is to utilize magnetic composites containing magnetically hard (HM) and soft (SM) phases that can benefit from the high coercivity of the HM phase and high saturation of the SM phase [13,14,15,16]. In this approach, ultra-high coercive alloys are especially interesting considering the fact that the increasing contribution of the SM phase usually leads to a significant decrease in the resulting coercivity of the whole composite [17,18]. We reported the possibility of obtaining bulk alloys of RE-Fe-Nb-B (RE = Tb,Dy) characterized by a more than 6 T coercive field at room temperature [19,20]. The key point for understanding this property lies in the specific microstructure of dendrite RE_2_Fe_14_B grains achieved by a proper Nb concentration in combination with a high cooling rate, through the use of the mold suction casting technique. Unfortunately, the magnetic saturation is not particularly high due to the antiferromagnetic Fe–(Tb,Dy) coupling. However, this type of bulk alloy is a good candidate for the so-called spring-exchange composites as a source of magnetic anisotropy. It was confirmed that partial substitution of Tb by non-magnetic Y results in the formation of a bulk composite containing Tb_2_Fe_14_B (HM) and Y_2_Fe_14_B (SM) compounds with a similar dendrite microstructure [21]. In this case, Y, with a similar atomic radius to Tb, plays the same role but is non-magnetic. The achievement was a significant increase in magnetic remanence (about twofold) and the energy product |*BH*|_max_ (about threefold). This suggested that further studies on other alloying additions for the RE–Fe–Nb–B type of alloys can broaden knowledge about their impacts on phase composition as well as microstructure, both useful in assisting in the design of new hard magnetic materials.

The aim of this paper was to study the influence of dissolving the (Fe_80_Nb_6_B_14_)_0.88_Dy_0.12_ alloys in zirconium on phase structure, microstructure, and magnetic properties. Zirconium was chosen because it has a different (from Dy) atomic radius and forms compounds other than 2-14-1 with Fe. Thus, we expected significant changes in phase composition as well as microstructure that both influence final magnetic properties.

## 2. Experimental Procedure

The initial alloy of (Fe_80_Nb_6_B_14_)_0.88_Dy_0.12_ was obtained using typical arc-melting in a chamber with an argon atmosphere using commercially available basic elements (purity of 99.9%). Next, the samples were crushed and mixed with a proper amount of Zr, following the formula [(Fe_80_Nb_6_B_14_)_0.88_Dy_0.12_]_1−x_Zr_x_ (*x* = 0; 0.01; 0.02; 0.05; 0.1; 0.2; 0.3; 0.5). Such compositions were melted and crushed several times in order to ensure the homogeneity of the ingredients. The final step involved the application of the vacuum mold suction (described in detail in [22]), leading to obtaining the samples in the form of rods 1 mm in diameter and 3 cm in length.

The phase compositions of the samples were determined through the X-ray diffraction (XRD) technique. The XRD measurements were carried out using a high-resolution PANalytical Empyrean diffractometer (Malvern Panalytical, Malvern, UK) with CuKα radiation (40 kV, 30 mA) equipped with a PIXcel detector (Malvern Panalytical, Malvern, UK). The diffraction patterns were collected using a 2θ scan from 10 to 100° with 0.0131° steps. The data analysis was carried out using HighScore Plus software supplied by PANalytical (Version 4.9, Malvern Panalytical, Malvern, UK). The ICDD PDF-4 database was used to identify the phases.

The microstructure of the examined alloys was studied using a scanning electron microscope (SEM) JEOL JSM7600F (JEOL Ltd., Tokio, Japan) with an X-ray micro-probe. The samples were added to a conductive resin and polished in the following procedure. Preliminary grinding was carried out using a polishing machine (Metkon, Forcipol 102, Mettkon Instrunents Inc., Bursa, Turkey) and water-based abrasive papers (Klingspor, (Klingspor AG, Haiger, Germany) gradation successively: 220, 360, 600, 800, 1000, 1200, 1500). Next, polishing with diamond suspensions (Buehler (Waukegan, IL, USA) gradation: 6, 3, 1 µm) was applied. The finishing polishing was performed using a polishing cloth (Buehler, Waukegan, IL, USA) and colloidal suspension of silica with a grain size of 0.04 µm (Struers, Cleveland, OH, USA). Pictures of the sample surface were collected using the back-scatter electron mode (BSE) showing contrasts of the differences in the mean atomic number between the individual points in the sample. The element maps were determined by means of the energy-dispersive X-ray spectrometry (EDX) option.

The magnetic measurements were carried out using the SQUID magnetometer (XL-7, Quantum Design, Quantum Desing, San Diego, CA, USA) in temperatures ranging from 2 K to 300 K and magnetic fields of ±7 T.

## 3. Results and Discussion

The phase structures of the examined alloys were determined using the XRD technique. As was expected, the relatively fast cooling rate during casting and the chemical composition of the alloys led to the formation of different binary and/or ternary compounds. Regarding the aim of this work, we focused on the phases that revealed magnetic properties. Figure 1 shows XRD patterns for all tested alloys and selected references for the most interesting magnetic phases. Table 1 quantitatively summarizes the contributions of these phases.

The alloys can be divided into two groups, i.e., containing (*x* < 0.05) and free (*x* ≥ 0.05) of the ternary Dy_2_Fe_14_B compounds that are the main hard magnetic phase. For *x* < 0.05, the contribution of this phase was high and was determined at the level of 96 wt.%. Additionally, a 3–4 wt.% content of DyFe_2_ (relatively soft magnetic) was detected. For the higher Zr concentration, the formation of Fe_2_Zr and Zr_3_Fe (*x* = 0.5) together with separations of Dy and Fe was observed. It was interesting that while keeping the Fe-Dy ratio constant, the increasing presence of Zr significantly changed the phase structure. However, it seems that the preferred formation of the Fe–Zr compounds leads to a relative deficiency of Fe for the synthesis of Dy_2_Fe_14_B. It should be emphasized that for *x* > 0.05, the alloys are a kind of bulk composite, containing internally coupled (interacting), different magnetic phases of crystalline Dy and other soft magnetic Fe-Zr compounds. This fact can have an important meaning due to the high magnetic moment of 9 µ_B_ (where µ_B_ is the Bohr magneton) attributed to the Dy atoms.

The microstructure of the studied alloys was determined by means of the SEM technique in the chemical contrast mode (back-scattered electrons). Figure 2 shows SEM pictures for selected alloy compositions. For the initial undiluted alloy (*x* = 0), the microstructure reveals micrometric dendrite grains that were expected and have been observed earlier elsewhere [19,20]. The presence of 2 wt.% of Zr caused partial fragmentation and smoothing of the grains, as shown in Figure 2b. The further increase in the *x* parameter up to 10 wt.% led to the formation of needle-like grains with micrometric and even submicrometric sizes (in one dimension, see Figure 2c,d). The higher dissolution degree of the initial alloy (i.e., for *x* ≥ 20) resulted in the growth of regular and relatively large grains.

In order to show chemical compositions of the visible areas, element maps of Fe, Dy, Zr, and Nb were determined using the SEM EDX technique. Figure 3 shows such element maps for the [(Fe_80_Nb_6_B_14_)_0.88_Dy_0.12_]_0.95_Zr_0.05_ alloy. Taking into consideration that the location of the EDX analysis is about 2 µm, it can be stated that the Fe, Dy, and Zr elements were situated mainly in the needle-like grains, while Nb could be found in the area outside of these grains.

The presented SEM observations lead to the conclusion that the dissolving of the (Fe_80_Nb_6_B_14)0.88_Dy_0.12_ alloy with zirconium results in remarkable changes of the microstructure. These changes should be discussed in terms of the phase structures determined by the XRD measurements. For *x* ≤ 2, the alloys contain mainly the Dy_2_Fe_14_B compound; however, the small amount of Zr influences crystal growth in the applied suction casting. Next, for *x* = 0.05, one can observe a needle-like microstructure of the mainly hexagonal Dy_2_Fe_17_ phase, which was detected only in this case. The higher Zr concentration causes the formation of grains relatively larger and more regular in shape than mainly cubic Fe-Zr compounds.

Magnetic properties of the examined alloys were studied using the SQUID magnetometer. In order to demonstrate the magnetization processes, a set of magnetic hysteresis loops was measured at different temperatures ranging from 2 K to 300 K. According to the phase structure changes (see Table 1), it is worth presenting the hysteresis separately for 0 ≤ *x* ≤ 0.05 and for 0.05 ≤ *x* ≤ 0.5. Figure 4 depicts full hysteresis loops for values of the *x* parameter up to 0.05, measured at 300 K. As shown, the hysteresis loop for the initial undiluted alloy was strongly asymmetric and revealed ultra-high coercivity of about 5 T (in the second quadrant). It is important to take note that the so-called reverse magnetization process occurs in two steps: the first in µ_0_*H* ≈ 0, and the second in µ_0_*H* ≈ −5 T. The low concentration of Zr (*x* = 0.01, *x* = 0.02) caused a decrease in the coercive field; additionally, the hysteresis loops were symmetric. The further increase in the Zr content (*x* = 0.05) resulted in a collapse of coercivity and a decrease in magnetic saturation. Figure 5 shows magnetic hysteresis loops for the alloys with *x* ≥ 0.05. A subsequent decrease in coercivity (with increasing *x*) and non-monotonic changes of magnetic saturation can be observed. The magnetic characteristics at lower temperatures (see Figure 6 for T = 2 K) are particularly interesting because magnetization processes are a competition between different kinds of energies, including magnetostatic, exchange, anisotropy, and thermal energies. In the case of the alloys containing the hard magnetic Dy_2_Fe_14_B phase, the low temperatures resulted in domination of the anisotropy energy and, applying the maximum external magnetic field available in our apparatus of ±7 T, the material could not be magnetically saturated. The hysteresis was strongly asymmetric, which apparently decreased the measured defined coercive fields. On the other hand, for the alloys containing separations of Dy (the Curie point equal to 92 K), one can expect a significant contribution of this element regarding its high atomic magnetic moment.

The values of coercive field *H*_c_, magnetic saturation *M*_s_, as well as magnetic remanence *M*_r_ for all tested temperatures are displayed in Table 2. For comparison, Figure 7 and Figure 8 (*H*_c_) and (*M*_s_ and *M*_r_) show these values as a function of the *x* parameter for T = 300 K and T = 2 K. Generally, the disappearance of the hard magnetic Dy_2_Fe_14_B phase caused the collapse of *H*_c_, especially at room temperature. However, it is quite surprising that the low-temperature measurements revealed non-zero (even more than 0.5 T) coercivity for the alloys free of the hard magnetic Dy_2_Fe_14_B and Dy_2_Fe_17_ phases, i.e., for *x* ≥ 0.1. Most likely, the anisotropy originated from the interphase regions where some Dy-Fe disordered phases can occur and influence magnetization processes via exchange interactions.

The changes of *M*_s_ and *M*_r_ as a function of the dissolution degree are also surprising. Following the increasing *x* parameter, 1 wt.% of Zr caused an increase in both parameters. Next, up to *x* = 0.1, one can observe a decrease in *M*_s_ and *M*_r_. The highest values of the analyzed quantities were noticed for *x* = 0.2 and *x* = 0.3. This effect can be explained based on the changing phase structure and the fact that magnetic moments of Dy and Fe in compounds are coupled antiferromagnetically. Exemplary for the Dy_2_Fe_14_B, the magnetic moment per formula unit can be estimated as 14 µ_Fe_−2µ _Dy_ ≈ 28 µ_B_−18 µ_B_−10 µ_B_. This means that the average magnetic moment calculated per Fe atom is 0.71 µ_B_. In contrast, for Fe_2_Zr, the average magnetic moment per Fe atom equals about 2 µ_B_. Therefore, the changing phase composition of the tested alloys can result in the observed increase in *M*_s_, even though increasing the *x* parameter means dissolution of the magnetic compound in non-magnetic Zr. Additionally, magnetic saturation and remanence are enhanced by the crystalline Dy when the temperature is below its Curie point.

Let us analyze the impact of the Zr addition for the alloys containing the hard magnetic Dy_2_Fe_14_B phase. Generally, the hysteresis loops for these alloys are not magnetically uniform, i.e., the reverse magnetization curve (the first and the second quadrant) reveals more than one maximum of d*M*/d*H* dependence. In the case of a material containing different magnetic phases/objects (uncoupled or not perfectly coupled) with different anisotropies, one can expect an occurrence of many magnetization speed maxima in different magnetic fields related to anisotropy fields of the phases. Figure 9 shows the d*M*/d*H* dependencies for *x* ≤ 0.05. In all cases, a soft magnetic phase is visible as a peak with the maximum placed at *H* = 0. For *x* = 0, there is a hard magnetic component with a maximum remagnetization speed of about 6 T. The addition of 1 wt.% of Zr leads to a shift of this maximum into lower fields as well as an appearance of the additional component at µ_0_*H* = 1.8 T. A more complex structure of magnetic objects reveals the alloys with 2 wt.% of Zr. Four components can be observed: soft magnetic at *H* = 0 as well as hard magnetic at 5.5 T, 4 T, and 1.8 T. According to this information, the phase structure is almost the same for *x* < 0.05, and the detected differences should be attributed to a changing distribution of Dy_2_Fe_14_B grains. This conclusion is in agreement with our previous simulation research, when a lowering of the anisotropy field with decreasing grain size was shown. In this variety of the *x* parameter, the SEM observations also indicated lowering of the grain size as a function of the increasing Zr addition.

In summary, it can be stated that the influence of dissolving (Fe_80_Nb_6_B_14_)_0.88_Dy_0.12_ alloys in zirconium is a complex problem. The Zr addition strongly affected the phase structure in the way that the formation of Fe–Zr compounds was preferred, and for *x* ≥ 0.05, the hard magnetic Dy_2_Fe_14_B compound disappeared. This hard magnetic phase was replaced by Fe_2_Zr and even Zr_3_Fe for the alloy with *x* = 0.5. The changes in the phase structure caused the observed appearance of crystalline Dy, it being a source of high magnetic moments at lower temperatures. In fact, this caused an interesting effect related to the magnetic state of Dy. At higher temperatures (above the Curie point of Dy), it was paramagnetic and uncoupled with the ferromagnetic Fe_2_Zr phase. Below this Curie point, one may expect the occurrence of the ferromagnetic exchange interaction between Dy atoms and antiferromagnetic coupling with Fe atoms in the interphase regions. In this situation, the magnetization processes are very complex because of the appearance of exchange interactions and exchange anisotropy. Finally, it is possible to “switch” on (or off) apparent coercivity and remanence of the Fe_2_Zr phase, which could be significant, for example, for sensor applications.

## 4. Conclusions

The aim of the presented work was to study structural and magnetic properties of the (Fe_80_Nb_6_B_14_)_0.88_Dy_0.12_ alloy diluted in non-magnetic Zr. The impact of the Zr addition can be divided into two groups of alloys, i.e., for *x* < 0.05 and for *x* ≥ 0.05. For the first group, the phase structure did not significantly change and revealed the occurrence of the dominant hard magnetic Dy_2_Fe_14_B compound as well as relatively soft DyFe_2_. However, it was shown that the observed fragmentation of the microstructure affected the magnetization processes of the examined alloys. The coercive field gradually decreased, while for *x* = 0.01, weak increases in *M*_s_ and *M*_r_ were detected. For *x* = 0.05, we observed a formation of needle-like micrometric grains of the mainly Dy_2_Fe_17_ phase. A further increase in Zr addition led to the increasing contribution of the Fe-Zr compounds and, simultaneously, the separation of crystalline Dy. In this range of the *x* parameter (i.e., for *x* = 0.2 and *x* = 0.3), significant increases in *M*_s_ and *M*_r_ were observed. Additionally, at lower temperatures, the hysteresis loops revealed the appearance of magnetic anisotropy originating from the ferromagnetic state of Dy and some Dy-Fe coupling in the interphase regions.

## Figures and Tables

**Figure 1 materials-14-02526-f001:**
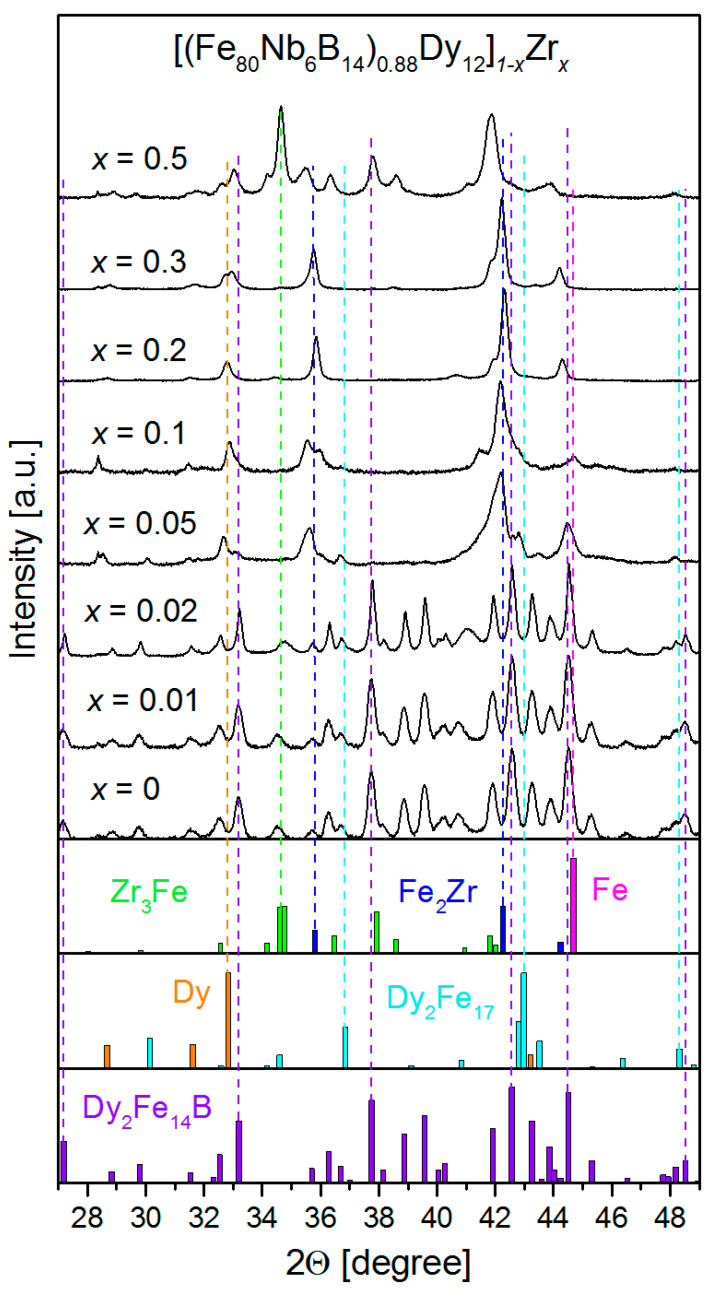
XRD patterns for all studied alloys and selected reference patterns for detected magnetic phases.

**Figure 2 materials-14-02526-f002:**
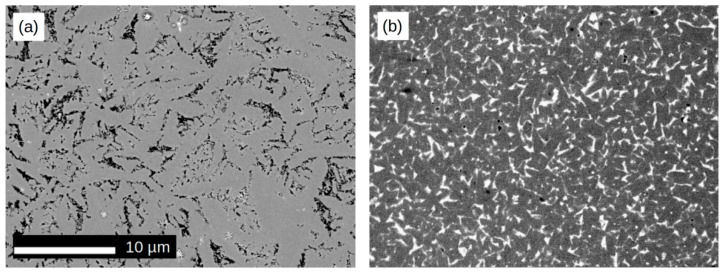
SEM images (BSE mode) for [(Fe_80_Nb_6_B_14_)_0.88_Dy_0.12_]_1−x_Zr_x_. (**a**) *x* = 0; (**b**) *x* = 0.02; (**c**) *x* = 0.05; (**d**) *x* = 0.1; (**e**) *x* = 0.2; (**f**) *x* = 0.3.

**Figure 3 materials-14-02526-f003:**
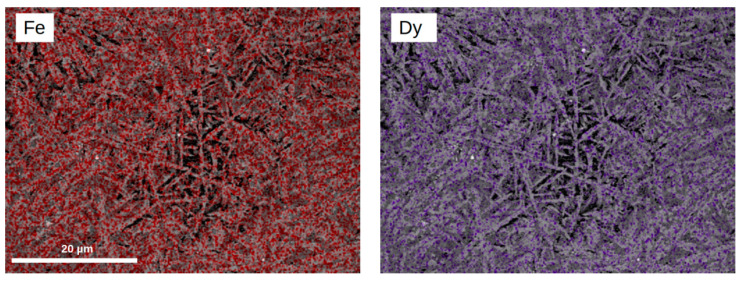
Maps of the selected elements for the [(Fe_80_Nb_6_B_14_)_0.88_Dy_0.12_]_0.95_Zr_0.05_ alloy determined using the EDX technique.

**Figure 4 materials-14-02526-f004:**
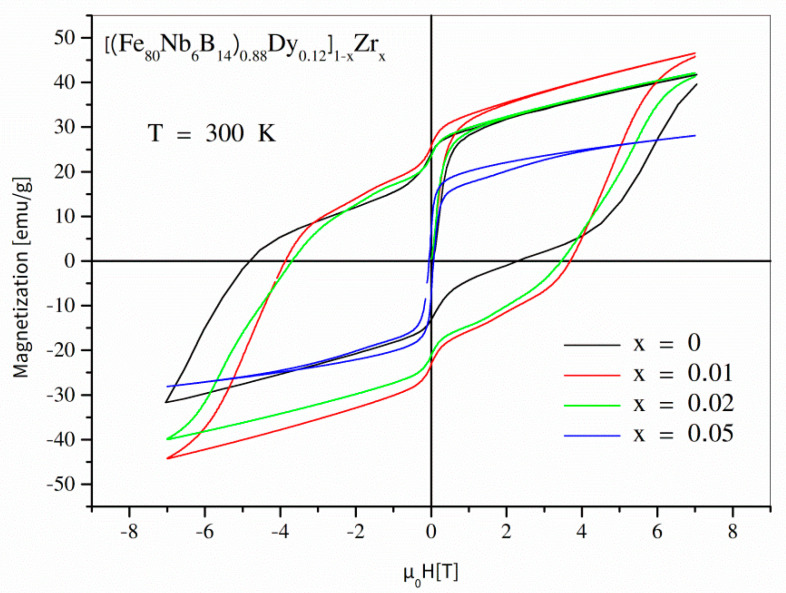
Hysteresis loops for the [(Fe_80_Nb_6_B_14_)_0.88_Dy_0.12_]_1−x_Zr_x_ (0 ≤ *x* ≤ 0.05) alloys measured at room temperature.

**Figure 5 materials-14-02526-f005:**
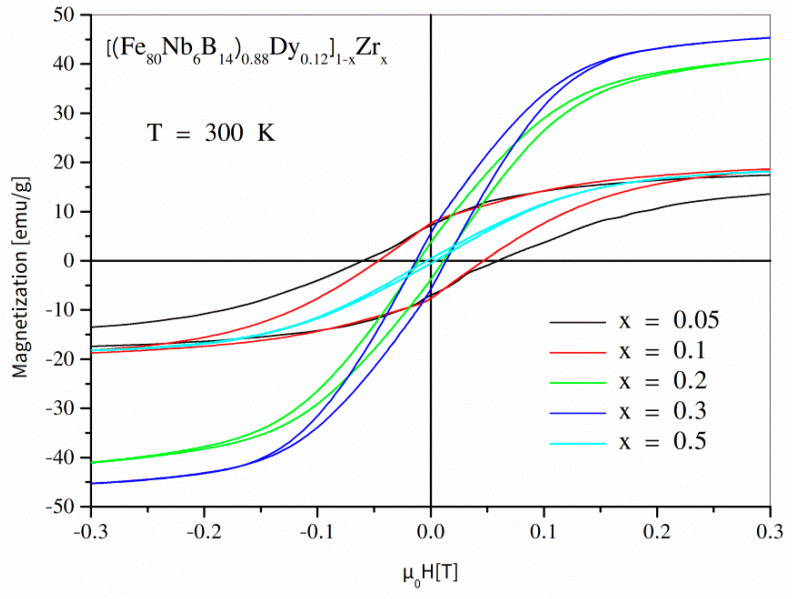
Hysteresis loops for the [(Fe_80_Nb_6_B_14_)_0.88_Dy_0.12_]_1−x_Zr_x_ (0.05 ≤ *x* ≤ 0.5) alloys measured at room temperature.

**Figure 6 materials-14-02526-f006:**
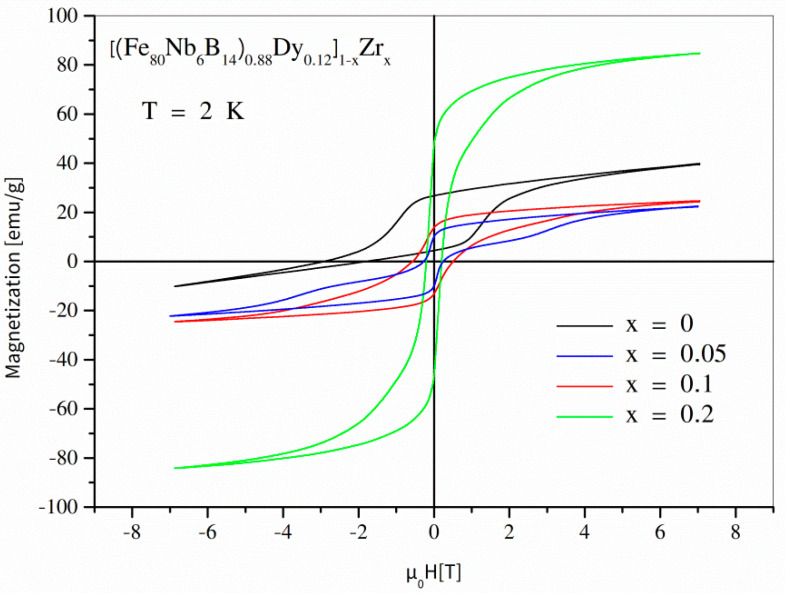
Hysteresis loops for selected [(Fe_80_Nb_6_B_14_)_0.88_Dy_0.12_]_1−x_Zr_x_ alloys measured at a low temperature of 2 K.

**Figure 7 materials-14-02526-f007:**
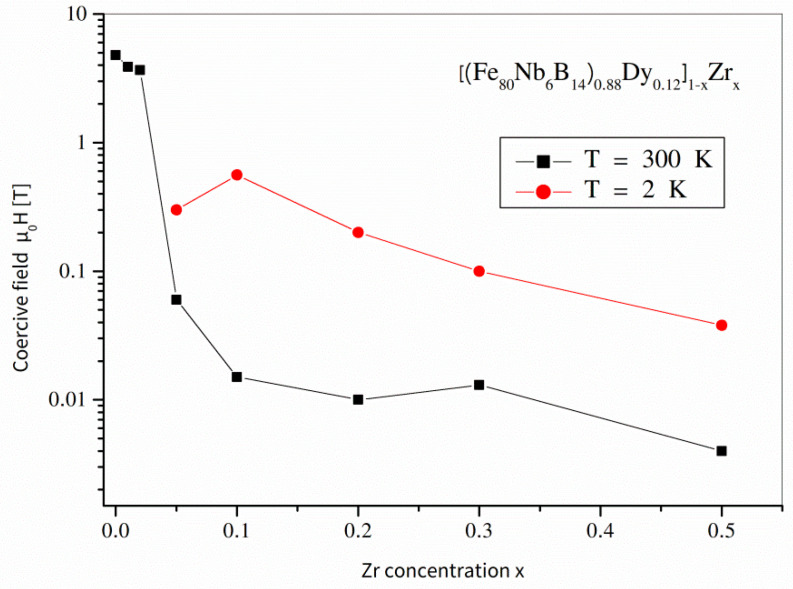
Coercive field *H*_c_ for the [(Fe_80_Nb_6_B_14_)_0.88_Dy_0.12_]_1−x_Zr_x_ (0 ≤ *x* ≤ 0.5) alloys determined at 300 K and 2 K.

**Figure 8 materials-14-02526-f008:**
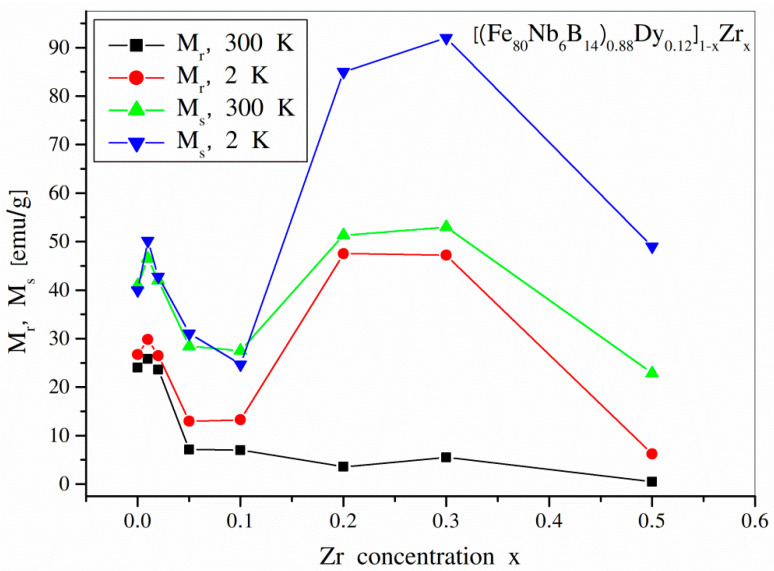
Magnetic saturation *M*_s_ and magnetic remanence *M*_r_ for the [(Fe_80_Nb_6_B_14_)_0.88_Dy_0.12_]_1−x_Zr_x_ (0 ≤ *x* ≤ 0.5) alloys determined at 300 K and 2 K.

**Figure 9 materials-14-02526-f009:**
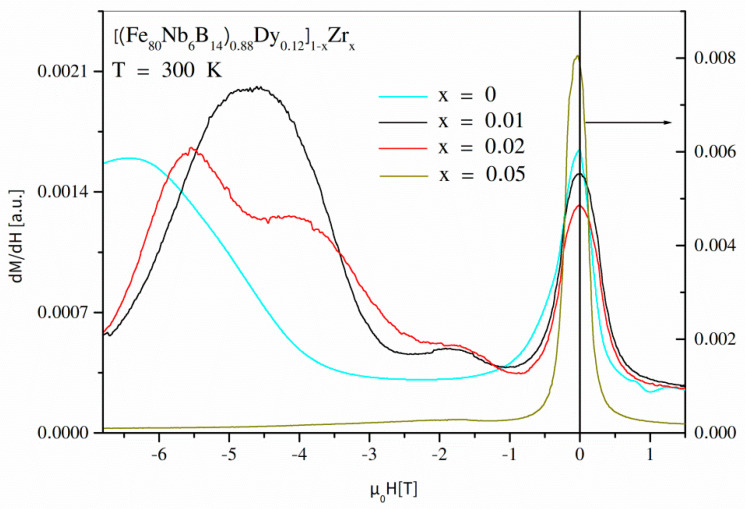
d*M*/d*H* dependencies for the [(Fe_80_Nb_6_B_14_)_0.88_Dy_0.12_]_1−x_Zr_x_ (0 ≤ *x* ≤ 0.5) alloys determined at 300 K.

**Table 1 materials-14-02526-t001:** Contributions of these phases (at.%) in [(Fe_80_Nb_6_B_14_)_0.88_Dy_0.12_]_1−x_Zr_x_ determined from the XRD analysis. The estimated error of the phase content is about 2%.

x	Dy_2_Fe_14_B	DyFe_2_	Dy_2_Fe_17_	Dy	Fe_2_Zr	Zr_3_Fe	Fe
0	96	4	-	-	-	-	-
0.01	96	4	-	-	-	-	-
0.02	96	3	-	-	-	-	-
0.05	-	-	23	-	15	-	14
0.1	-	-	-	9	28	-	26
0.2	-	4	-	10	76	-	-
0.3	-	2	-	7	78	-	-
0.5	-	-	-	7	11	60	3

**Table 2 materials-14-02526-t002:** Coercive field *H*_c_, magnetic saturation *M*_s_ as well as magnetic remanence *M*_r_ for all tested alloys determined from hysteresis loops measured at different temperatures. The errors of the listed values are in the level of the last printed digit.

*x*	µ_0_*H*_c_ (T)					*M*_s_ (emu/g)					*M*_r_ (emu/g)				
	300 K	200 K	100 K	10 K	2 K	300 K	200 K	100 K	10 K	2 K	300 K	200 K	100 K	10 K	2 K
0	4.79	-	-	-	-	41.08	38.40	38.83	39.85	39.97	24.15	22.90	25.44	26.78	27.10
0.01	3.83	5.56	-	-	-	46.12	43.02	42.12	42.66	42.82	25.86	24.46	24.95	27.60	29.84
0.02	3.70	5.19	-	-	-	42.15	39.30	38.45	28.2	38.04	23.62	22.19	23.09	25.24	24.95
0.05	0.06	0.07	0.13	0.27	0.31	28.14	25.38	25.22	23.37	22.28	7.16	6.85	7.74	10.03	11.18
0.1	0.04	0.1	0.19	0.56	0.54	27.56	23.46	22.96	25.04	24.87	8.12	8.65	9.19	13.70	13.75
0.2	0.01	0.02	0.05	0.16	0.20	51.41	63.34	79.50	84.78	84.92	3.56	8.53	19.34	43.35	47.38
0.3	0.01	0.02	0.04	0.09	0.10	53.17	64.32	83.25	91.54	92.16	5.76	10.28	20.47	44.71	46.94
0.5	0.004	0.005	0.012	0.031	0.038	22.89	29.31	41.43	48.83	49.13	0.53	0.79	1.88	5.15	6.20

## Data Availability

Data are available on request from the corresponding author.

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
