# Peer review of "Influence of Dissolving Fe–Nb–B–Dy Alloys in Zirconium on Phase Structure, Microstructure and Magnetic Properties"

_materials, 2021, doi:10.3390/ma14102526_

Round 1

Reviewer 1 Report

referee report 
materials-1204843
Influence of dissolving the Fe-Nb-B-Dy alloys in zirconium on phase structure, microstructure and magnetic properties
Grzegorz Ziólkowski, Artur Chrobak, Ewa Talik, Joann Klimontko and Dariusz Chrobak

This manuscript reports on the effect when dissolving the Fe-Nb-B-Dy alloys in zirconium on phase structure, 
microstructure and the resulting magnetic properties. This is an important issue in current research trying to reduce
the amount of reare earths in the fabrication of hard magnetic materials. Thus, the topic is very interesting for Materials.

The manuscript is overall well arranged, comprises 9 figures and 22 references. Two obvious problems are the English
language, so please ask a native speaker for help, and several typos like "filed" (Introduction).
Another technical problem is that there should always be space between a physical quantity and its unit, and please also
check the spaces between text and parentheses. The unit for Kelvin is aways a capital "K".

Some other points:
# Please also use proper spaces for all formulae in the text.
# All microscopic figures require readable text -- this is in all cases too small.
# Please give details of the sample polishing you have applied -- "smooth surface" is not scientific at all.

Overall, this manuscript may be published in Materials after a major revision.

Reviewer 2 Report

The manuscript shows and discusses the effects of diluting (FeNbB)Dy alloy in Zr with reference to two, linked, effects: the formation of different phases (as a function of the fraction x of Zr in each phase), and the (consequent) influence on the magnetic processes (hysteresis cycle parameters: coercivity and saturation magnetization). The research is definitely high level, and exploits appropriate and advanced technology to fabricate the systems (crushing and arc-melting followed by vacuum mold suction), characterize them by structural analysis (X-ray diffraction, scanning electron microscopy with backscattered electrons, space-resolved element-map by energy-dispersive-X-ray-spectrometry), magnetic characterization (by SQUID magnetometry in temperature range 2 K / 300 K, and applied magnetic fields up to 7 T). The study shows the formation of two regimes depending a limit-concentration of Zr (specifically, x=0.05). The discussion of section 3 is in my opinion particularly clear and convincing, with reference to the helpful figures and tables, even concerning the unexpected results of Fig. 8, interpreted through the antiferromagnetic coupling. Even though this kind of study is intrinsically difficult, definite regularities were spotted, analyzed and justified, which are of great help and interest for the scientific community. Hence, I definitely recommend publication of the manuscript. Only, I suggest to doublecheck a few misprints, among which:

  1. 2, sec. 3, line 1: “the examined alloys” add ‘d’
  2. 4, line 6: “…leads to a relative deficiency” I presume (not “deficient”)
  3. Figure 2: add “(d)” label into the corresponding inset (now missing)
  4. Figure 3: caption reads mistakenly “Figure 2”
  5. 8 line 1: “…collecting and ‘plotting’ the values”
  6. 11 afte “In summary”, 5 lines below: “…Zr contribution reaches 50%. Moreover, the free…” tese two linked(?) sentences are definitely not clear.
  7. In “Conclusions”, at line 8: “…weak increase”

Round 2

Reviewer 1 Report

The authors have considered most of the points mentioned by the refereees and corrected the obvious mistakes. However, the authors did not take care to improve their descriptions of the methods. I consider the comment the explanation about "smooth surface" has been specified, i.e., “smooth surface on nanometer scale" as a mere joke. What scientific value has this statement? It would be important here to explain how you did polish the sample surface -- which method, what materials, etc. This would help the reader to understand the problems you may have in the interpretation of your data.

So, please provide a proper revision -- the present one is simply insufficient.
